# How Does Liquid-Liquid Phase Separation in Model Membranes Reflect Cell Membrane Heterogeneity?

**DOI:** 10.3390/membranes11050323

**Published:** 2021-04-28

**Authors:** Taras Sych, Cenk Onur Gurdap, Linda Wedemann, Erdinc Sezgin

**Affiliations:** Science for Life Laboratory, Department of Women’s and Children’s Health, Karolinska Institutet, 17165 Solna, Sweden; taras.sych@ki.se (T.S.); gurdap@kth.se (C.O.G.); wedemann@kth.se (L.W.)

**Keywords:** phase separation, liquid-ordered domains, liquid-disordered domains, GUVs, GPMVs, plasma membrane, lipid rafts, plasma membrane vesicles, SLBs

## Abstract

Although liquid–liquid phase separation of cytoplasmic or nuclear components in cells has been a major focus in cell biology, it is only recently that the principle of phase separation has been a long-standing concept and extensively studied in biomembranes. Membrane phase separation has been reconstituted in simplified model systems, and its detailed physicochemical principles, including essential phase diagrams, have been extensively explored. These model membrane systems have proven very useful to study the heterogeneity in cellular membranes, however, concerns have been raised about how reliably they can represent native membranes. In this review, we will discuss how phase-separated membrane systems can mimic cellular membranes and where they fail to reflect the native cell membrane heterogeneity. We also include a few humble suggestions on which phase-separated systems should be used for certain applications, and which interpretations should be avoided to prevent unreliable conclusions.

## 1. Introduction

Membranes are complex structures comprised primarily of lipids and proteins. Although membranes have been a major research focus for over a century [1], the exact lipid and protein composition of plasma membrane (PM) is still under investigation. PM executes a diversity of molecular functions, many of which are cell type specific and require specific lipid environments, which leads to diversity in composition between different types of cells [2]. Integrally embedded or peripherally associated proteins further increase the complexity in the membrane environment. Hundreds of different lipids and membrane proteins have been identified, and our understanding of their function is continuously rising [3,4].

This incredible complexity of PM constitutes one of the most fundamental questions in cell biology: **Why do we have so many lipids**? Evidence shows that each lipid type contributes differently to physical properties and cellular processes in the PM. Certain lipids, such as ceramides, lysolipids, and conical phospholipids are known to induce curvature essential for endo- and exocytosis [5]. Some lipids, such as cholesterol, ensure the essential elasticity of the membrane while some others are directly involved in signaling [6,7]. Therefore, lipid composition in cells is spatially and temporally regulated. One of such regulation mechanisms is PM asymmetry. Asymmetry between the lipid bilayer leaflets has been shown since the 1970s with the identification of sphingomyelin (SM) and phosphatidylcholine (PC) mainly in outer leaflet, and phosphatidylethanolamine (PE) and -serine (PS) in inner leaflet [8]. Recent lipidomics investigations additionally highlighted asymmetry in the acyl chain saturation between the leaflets [9]. Saturated acyl acids are predominantly found in the outer leaflet, while that of cytoplasm mostly harbors unsaturated acyl chain lipids [9].

Another important principle for functional regulation of PM components is lateral heterogeneity via formation of nanodomains or nanoclusters. Favorable interactions between SM and cholesterol have been known since the 1970s and, together with other biochemical and biophysical evidence, led to the “lipid raft” model [10]. This model predicts that the interaction of cholesterol with saturated lipids, especially SM, leads to the formation of nanodomains within the PM. SM and cholesterol-rich domains, termed “lipid rafts”, are more ordered and densely packed than the surrounding “sea” of unsaturated fatty acids. Unsaturated membranes are less densely packed due to the space requirement of the unsaturated acyl chain and hence are more fluid.

This model of lipid rafts as dynamic, sphingolipid/sterol-enriched domains [11] has been intensely investigated. After the postulation of this concept, it was thought to solve every question in cell biology; as such, with a slight exaggeration, almost every membrane protein was found to be associated with lipid rafts. The protein association in lipid rafts also led to a revised term: “membrane rafts” [12], which involved not only lipids but also proteins in the concept. Soon enough though, thanks to critical experiments in the following years, the main methodology to determine raft association of proteins, namely detergent resistance assay, was found to be unreliable and largely nonreproducible. Although this was a major setback for the raft concept, biophysical investigations of the core idea of lateral membrane heterogeneity did not vanish (unlike biologists’ enthusiasm to find their proteins in lipid rafts). These biophysical experiments enlightened the physical chemistry of the concept of “phase separation” as well as the roles of lipids in the physics of membranes. Seminal work during this time includes the formation of model membranes exhibiting liquid–liquid phase separation (LLPS) [13], composed of the ternary mixtures of unsaturated lipid 1,2-Dioleoyl-sn-glycero-3-phosphocholine (DOPC), sphingomyelin (SM), and cholesterol (Chol), and determining the phase diagram of these ternary lipid mixtures [14,15,16]. Furthermore, additional evidence was obtained on why and when microscopic phase separation can or cannot exist in cell membranes, for example, by experiments showing the possible interaction of cholesterol with gangliosides [17] or the impact of cortical actin cytoskeleton [18] on lateral membrane heterogeneity.

Despite the frequently used humour on how “rafts get smaller as the resolution of the microscopes gets better”, indeed super-resolution microscopy immensely helped our understanding of the PM heterogeneity. Seminal work by Katharina Gaus and co-workers showed strong evidence for nanoscale domains in the PM [19]. Lipid–lipid interactions that lead to domain formation were clearly shown in mass spectroscopy approaches [17,20,21]. With the advances in lipidomics, a new age for membrane biology started. Currently, it is undeniable that PM is an immensely heterogenous structure full of nanoscale clusters and domains. However, source principal of this heterogeneity is still controversial. Is it lipid–lipid interactions, protein–lipid interactions, or do lipids actively participate in domain formation or passively follow the protein organization [22,23,24]?

To address these questions in highly complex PM is extremely challenging. Therefore, model membrane systems with reduced complexity are extensively used. These model membrane systems can mimic different aspects of membrane structure and dynamics. Commonly used model membrane systems are supported lipid bilayer systems (SLBs), large unilamellar vesicles (LUVs), giant unilamellar vesicles (GUVs), and cell-derived giant plasma membrane vesicles (GPMVs). Vesicles are freestanding systems while SLBs can either be supported by a planar or a spherical substrate. These different model membranes have their advantages and caveats, therefore, the choice of the model system and awareness of the limitations of the system are required to obtain translatable results to the natural PM [25]. LLPS can be reconstituted in all these model systems, yet the phases in different model systems are quite different in terms of their biophysical properties.

In the next section, we will discuss the phase separation in these model systems and the discrepancies in their biophysical properties.

## 2. Phase Separation in Model Membranes

Eukaryotic cells spend a considerable amount of resources to generate hundreds of various lipid species, leading to numerous lipid interactions. Different interaction affinities can potentially create membrane heterogeneity [26,27]. Unique interactions between saturated lipids and cholesterol result in the tighter-packed and higher-ordered membrane environment that segregates from relatively less organized and more fluid membrane regions consisting of unsaturated lipids [10,28]. The compositional and topographical complexity of PM makes it challenging to study specific lipid–lipid interactions [29]. This challenge has facilitated the development of model membrane systems that faithfully control molecular complexity to unravel the biophysical principles of lipid–lipid and protein–lipid interactions [30], and consequent formation of segregated domains, clusters, and molecular complexes [31]. Liquid–liquid phase separation in membranes is one of the key consequences of the aforementioned lipid–lipid interaction and leads to heterogeneity in lipid lateral diffusion [25,32], lipid packing and viscosity [33,34]. Yet, in live cells in their non-equilibrium state, observation and investigation of such phase separation remain challenging. Thus, membrane models featuring LLPS in thermodynamic equilibrium have been intensively used to examine phase separation in the membrane [32]. When certain proportions of cholesterol, saturated lipids (which have high melting temperatures), and unsaturated lipids (which have low melting temperatures) are mixed in vitro at appropriate temperatures, liquid-disordered (Ld) and liquid-ordered (Lo) domains can coexist. Lo phase is generally thicker, more ordered, more viscous, and has lower diffusivity compared to Ld phase.

The most widely used phase-separated model membranes are SLBs, GUVs, and GPMVs. Although there are also several key applications of other membrane systems, such as phase-separated small liposomes, for the scope of this review, we will focus on these three model systems. SLBs were established as a flat model membrane with a very thin layer of water between the membrane and its support [35]. Thanks to that, lateral and rotational movement of lipids as well as structural properties of the bilayer can be preserved, providing a controlled environment for dynamics studies [36]. However, there are also observations of membrane interaction with the support and its impact on the molecular movement in the membrane [25,37]. Obviously, the material of the substrate as well as the methodology of substrate treatment and SLB preparation define parameters of membrane coupling to its support. For example, lipid mobility in SLBs is different for several widely used support materials such as mica (higher mobility), glass, and silica (lower mobility) [25,38]. Moreover, the dynamics of ordered and disordered domain formation in phase-separated SLBs is also influenced by the substrate material [39,40]. Besides the ones mentioned before, several bio-compatible polymer-based SLB substrate materials were developed [41,42]. Furthermore, the deposition of the lipid bilayer on a porous surface allows for the production of so-called pore-spanning SLBs that offer the possibility to study constrained planar regions of free-standing membranes [43].

GUVs, in contrast, are produced as spherical free-standing bilayers [44] and are more similar to live cells in structure, geometry, and size, which makes them a useful model to investigate individual lipid types [45] and LLPS [46,47]. Yet, they are more sensitive to harsh experimental conditions [48]. Furthermore, although complex lipid compositions can be designed to some extent with GUVs, it is still challenging to reach a high level of lipid and protein complexity [30]. Instead, GPMVs—also free-standing—can be harvested from living cells, and they retain the compositional lipid and protein complexity of the native cell to a large extent [49]. Thus, in the studies of membrane structure and dynamics, GPMVs have shown to be an effective model [50].

Physical principles underlying LLPS and the contribution of different lipids and compounds to LLPS have been extensively investigated using phase-separated model membrane systems [51,52,53,54]. The necessity for LLPS in a lipid bilayer would be the presence of lipid species with different chain-melting temperatures [55]. Additionally, lipid chain length mismatch contributes to the LLPS [56,57]. Notably, the geometrical and biophysical properties can be modulated by tuning parameters such as lipid composition, temperature, osmotic pressure, membrane tension, etc. [58] As a result, in synthetic membranes (e.g., GUVs and SLBs) composed of the ternary mixture (e.g., of DOPC, SM and Chol, classical raft mixture), LLPS exists at specific lipid ratios [15,16]. Moreover, in extreme cases, three-phase coexistence can be observed, where in addition to Lo and Ld, “gel phase” domains appear [14,59].

In addition, the ability of biological stimuli to induce LLPS [60] has been used to unravel the effect of sterols or lipid-binding compounds such as cholera toxin [61].

Functional protein reconstitution in phase-separated membrane models was a crucial step in gaining a better understanding on protein behavior in phase-separated model systems and their association with nanodomains [62,63]. Protein composition in GUVs or SLBs is a major limitation which was overcome by GPMVs derived directly from cells. It was shown that certain proteins prefer ordered phase while others prefer disordered phase in GPMVs [49]. This differential partitioning of proteins was recently attributed to different transmembrane domain sequences and acylation of proteins [64]. There is still ongoing effort to understand how proteins can segregate into phases, and there are still several open questions: for instance, how molecular interactions or clustering of proteins influence their partitioning, whether intracellular or extracellular domains have roles in their partitioning, or whether each protein has a distinct lipid environment surrounding them.

Although these model systems provided critical knowledge on membrane biophysics, there is still a fundamental question: How well can they mimic the native cellular processes? Before discussing this, it is useful to understand the main technologies we use to visualize phase separation in biomembranes.

## 3. Fluorescence to Study Phase Separation

There are several biophysical methods to study phase separation, such as atomic force microscopy, Cryo-Electron Microscopy, Nuclear Magnetic Resonance, Surface Plasmon Resonance, Raman spectroscopy, mass spectroscopy, etc. Here, we will focus on fluorescence microscopy. The visualization of phase separation in lipid membranes with fluorescence microscopy usually requires insertion of lipophilic fluorescent probes into the lipid bilayer. Over the last decades, a huge variety of molecular probes has been developed, including phase-selective lipids modified with fluorescent moieties and phase-sensitive fluorescent dyes [65].

*Phase-selective membrane probes:* The obvious candidates to visualize phase separation would be lipids with a strong preference to liquid-ordered or liquid-disordered membrane domains. Fluorescently-modified lipids with unsaturated fatty acyl chains (e.g., oleoyl lipids) are widely used as Ld phase markers whereas lipids with saturated chains (palmitoyl or stearoyl lipids, ceramide, sphingolipids) as well as cholesterol can potentially be employed to mark Lo phase domains [66]. While unsaturated lipid analogues often prefer Ld phase as expected, Lo partitioning is significantly harder to achieve. Addition of a fluorescent moiety either to the fatty acyl chain or to the head group renders the resulting construct bulkier, hence lipids are likely to lose their preference to dense Lo domains and instead incorporate in Ld. Thus, the membrane probes based on phospholipids with saturated fatty acyl chains (e.g., based on palmitic or stearic acids) such as DPPE-Rhodamine [66], DSPE-Abberior Star Red (DSPE-ASR) [67], DHPE-TexasRed [68], and many others incorporate preferentially to Ld. Moreover, the majority of existing fluorescent sphingomyelin analogues vary in the length of their fatty acyl chain partition to Ld in GUVs and GPMVs [32,66]. However, at least one known specie (5-Bodipy-SM) partitions in Ld in GUVs, but in Lo, partitions in GPMVs [32]. On the contrary, several fluorescent cholesterol derivatives (e.g., TopFluor-Cholesterol, Cholestatrienol) do not lose their preference to Lo [32,66].

Besides changing the size and packing of the lipids, fluorescent tags can interact directly with the membrane, thus skewing the partitioning coefficient of the lipids toward Ld phase [69]. The preference of headgroup-modified lipids to Lo can be recovered by the introduction of a long hydrophilic spacer (e.g., polyethylene glycol—PEG) between the headgroup and fluorescent moiety that blocks the interactions of the dyes with the membrane. As an example, DSPE-Abberior Star Red partitions preferentially to Ld whereas DSPE-PEG-Abberior Star Red partitions to Lo (Figure 1) [67]. Furthermore, liquid phases can be visualized by proteins (such as lysenin, specific to sphingomyelin [70]) and charged peptides (such as daptomycin, specific to PG [71]) that specifically bind membrane lipids.

Glycosphingolipids (GSLs) are known to associate with sphingolipids and cholesterol, and they can be employed as markers of ordered domains [17,72]. In synthetic and cell-derived membrane systems, glycosphingolipids with saturated fatty acyl chains incorporate in Lo, whereas GSLs with unsaturated chains prefer Ld domains. This was shown for GSLs monosialotetrahexosylganglioside (GM1), and globotriaosylceramide (Gb3) [32,68,73,74,75]. Furthermore, introduction of a fluorophore can induce the loss of specificity to Lo domains, and similar to the phospholipids, introduction of a flexible spacer between the fluorescent moiety and GSL headgroup can improve partitioning to Lo [68]. The oligosaccharide moieties of GSLs can be specifically bound by carbohydrate-binding proteins (lectins). Hence, lectins can be employed to recognize GSLs, and as the membrane domains that GSLs reside [75]. However, such an approach requires caution as lectins not only bind but also cluster at the lipid bilayer surface, inducing membrane reorganization [55]. In particular, lectins such as Cholera toxin from *V. cholerae* (specific to GM1) and Shiga toxin from *S. dysenteriae* (specific to Gb3) can trigger phase separation in non-phase separated model membranes [76,77], induce reorganization of existing Lo domains [78], and sorting of GSLs [79].

Developing probes for visualization of Lo and Ld phases in phase-separated systems is still an important venue in chemical biology. Better probes are being constantly produced while the old ones are being better characterized. In our opinion, the best probes are biologically-inert molecules that would show a high partitioning coefficient (k_p,Lo_ ≥ 2, see Figure 1). Inertness rules out partitioning bias due to molecular interactions with other molecules while high partitioning coefficients will clearly distinguish phases, even when the packing difference between Lo and Ld phase is marginal. Although there is no rule of thumb for generating phase-selective lipid analogues, there are a few established observations on labelling strategies. Small fluorescent tags are usually better than large ones. Hydrophobic tags tend to interact with the membrane; therefore, they should be avoided for lipid labeling [69]. Acyl chain labeling usually interferes with the native behavior of the lipids, yet head group labeling is not a given success, and it might also affect the native lipid behavior. Inserting an inert linker between the head group and the fluorescent tag might help restore some properties while some others (such as head group interactions) might be affected. For all these reasons, new probes should be characterized properly and carefully. One should not assume that a saturated lipid analogue will report on ordered domains in cells or will be a representative of its native counterpart.

*Phase-sensitive membrane probes:* Numerous fluorescent dyes were synthetically produced and employed as lipophilic membrane markers [66,80,81]. Among them, environmental-sensitive membrane probes comprise small lipophilic molecules that typically incorporate efficiently in both Ld and Lo phases [65]. However, they exhibit different photophysical properties in different lipid environments. Widely used are membrane probes based on the dyes sensitive to the polarity of the environment [82,83,84,85,86,87]. Such molecules exhibit a red shift in their fluorescence emission spectrum in more polar environments, due to the so-called “solvent relaxation” [88]. Ld domains are more fluid and less tightly packed, hence the environment is more polar due to water penetration in comparison to Lo domains [89]. As a result, the fluorescence emission spectrum of environmental sensitive probe in Ld phase exhibits a drastic red shift in comparison to the probe in Lo phase (Figure 2A,B). Quantitatively, such red shift can be expressed by calculating the “general polarization” (*GP*) of the fluorescence emission spectrum:(1)$$GP=\frac{I_{blue}-I_{red}}{I_{blue}+I_{red}}$$
where *I_blue_* and *I_red_* are fluorescence intensities of short-wavelength part (“blue”) and long-wavelength part (“red”) of the emission spectrum of membrane marker, respectively. GP ranges from −1 to 1, and depends inversely on environment polarity (i.e., numerically increases with membrane rigidity). This allows not only for visualization of ordered and disordered domains but also for quantitative assessment of packing order in lipid bilayers of different compositions. This approach is widely used for quantitative comparison of the membrane order in the Lo and Ld phases in phase-separated model membranes [15].

New, better, more sensitive, and more photostable environmental-sensitive probes are constantly being produced. The gold standard for sensitivity is still one of the oldest probes, Laurdan. However, recently, a Prodan derivative (Pro12A) was shown to have better sensitivity for molecular order [85] while labeling only the outer leaflet of the PM. Other probes based on Nile Red [84,90,91] or Di-4-ANEP [19] are also commonly used for this purpose.

Fluorescent molecules that undergo excited state reactions (e.g., excited state intramolecular proton transfer [92,93]) or molecular rotors that undergo cis-trans isomerization in the excited state [94,95] can be employed as the sensors of membrane viscosity. Fluorescence lifetime of such probes increases with the viscosity of the environment. Hence, Fluorescence Lifetime Imaging Microscopy (FLIM) is applicable to visualize ordered and disordered domains in phase-separated membrane systems labeled with such molecules [34,96,97,98,99].

## 4. Biophysical Properties of Phases and Their Implications

Caveats of model membranes were reviewed in detail in ref [100], but briefly stated, each model has its own artifacts, weaknesses, and challenges. One should be aware of these limitations to avoid misinterpretations. When phase separation is investigated, the biophysical properties of domains, such as molecular packing, are a crucial factor to consider. It is absolutely critical to remember that different lipid compositions lead to different molecular packing [101]. Similarly, phases for different model systems are not identical [32,102]. It was demonstrated previously [32] that membrane order of phases in GUVs composed of the commonly used mixture (DOPC:SM:Chol) is extreme; that is, Lo phase is extremely ordered, and Ld phase is extremely disordered [32,101,102] (Figure 2A,B). The natural implication of the extreme order of Lo phase is significant exclusion of many molecules from Lo domains because they cannot easily fit into the tightly-packed membranes (Figure 2C). Therefore, despite being an excellent tool to study the phenomenon of phase separation and determine its physical foundations, phase-separated GUVs composed of the commonly used ternary mixture of DOPC, SM and Chol are not good mimics of any possible heterogeneity in PM. For this purpose, GPMVs serve as better systems since they are cell-derived, and they preserve natural complexity to a large extent. It was shown that molecular order of phases in GPMVs is only moderately different; that is, ordered domain is only moderately more ordered than disordered domain [32,103] (Figure 2B). This non-extremity makes GPMVs a good platform to study protein or lipid partitioning. Notably, phase partitioning preference of several molecules differs in GUVs of classical mixture and GPMVs due to this discrepancy. For instance, several saturated lipid analogs were shown to prefer Ld phase in DOPC:SM:Chol or DOPC:DPPC:Chol GUVs while they partition preferentially in Lo phase in GPMVs (Figure 2C). Similarly, a bona fide ordered phase marker of GPI-anchored proteins partitioned into Ld phase when reconstituted in phase-separated GUVs [104].

It is important to note that the order of liquid phases in GUVs can be modulated by tuning the lipid composition and introduction of various lipid species into the mixture. For instance, replacing DOPC with POPC in lipid mixtures increases the lipid order of the Ld domains, making it more similar to Ld domains in GPMV membrane [105]. Furthermore, introducing different sphingolipid species [59] or saturated phospholipids such as DSPC or DPPC [105,106] to the mixture modulates morphology and order in Lo domains.

In general, it is not wise to draw direct conclusions on cell membrane heterogeneity based on phase-separated model membranes owing to their thermodynamical equilibrium. It is, however, particularly dangerous to rely on partitioning in GUVs with extreme order differences in Lo and Ld phases, especially for molecules that show Ld partitioning in GUVs. This could simply be due to the extremely tightly-packed Lo phase of GUVs. For this reason, if the partitioning of a biological molecule is investigated, GPMVs will give more reliable results. However, GPMVs are no saints; for instance, they lack an organized cortical actin cytoskeleton which is a major contributor to membrane structure [107]. Moreover, membrane asymmetry is partially broken in GPMVs, making it difficult to study leaflet biology. Lipid and protein composition of GPMVs are also not exactly the same as in live cell PM [108]. Lastly, they are usually obtained using vesiculating agents whose effects on the membrane lipids and proteins are not fully understood. As far as is known, one of these such agents, dithiothreitol (DTT), can break thioester bonds; thus, palmitoylated proteins cannot be studied using GPMVs prepared with DTT. Instead, other vesiculating agents such as N-ethylmaleimide (NEM) should be used for protein studies.

## 5. LLPS in Model Membranes vs. Live Cell PM Heterogeneity

So far, we have focused on microscopic phase separation in model membrane systems. In cells, microscopic phase separation does not naturally occur. Instead, live cell membrane heterogeneity exists at the nanoscale that requires more delicate technologies. For example, super-resolution microscopy has been extensively applied to elucidate the nanoscale PM structure and dynamics [109,110,111].

Another major difference between live cell PM heterogeneity and LLPS in model membranes originates from the absence of live cell dynamics in the latter. LLPS in model membranes is usually studied at thermodynamic equilibrium whereas the cell PM is far from such conditions. Moreover, temperature is a crucial parameter for cellular homeostasis, and LLPS experiments are usually performed at temperatures lower than physiological temperature (e.g., room temperature), which biases the conclusions. Finally, on top of PM’s own complexity, there is a complex interplay between PM and other cellular elements such as cytoskeleton and glycocalyx that are usually missing in model membranes.

Despite these shortcomings, LLPS in model membranes is an important tool to give insight into the partitioning propensity of molecules, that is, whether they prefer relatively more fluid or more rigid lipid environments. These collective lipid–protein interactions might be key for compartmentalizing the signaling in cells. Phase separation is not limited to the PM; recent evidence shows that intracellular organelles can also exhibit phase separation that might be crucial for organelle function [112,113,114]. Thus, understanding phase separation will help us reveal the physical principles underlying compartmentalized cellular life. Phase separation in model membranes is a unique way to achieve this. Besides “collective” lipid–protein compartmentalization, specific interactions between macromolecules also contribute to the regulation of cellular signaling. To study the role of individual components in cells usually requires elimination of the molecules of interest. While it is trivial to knockout proteins in live cell context, it is not so trivial to achieve this for lipids. Enzymes involved in lipid synthesis are often multi-tasking, that is, one enzyme is involved in the synthesis of multiple lipids. Therefore, despite their limitations mentioned above, model membranes are unique tools to discern specific lipid–lipid or lipid–protein interactions owing to their tightly controllable compositions.

## 6. Conclusions

In this review, we tried to cover how phase separation in model systems can be studied, and how well they do or do not reflect the real cell membrane heterogeneity.

In spite of being studied for decades, phase separation continues to amaze scientists. The new concept of “membraneless organelles” is based on phase separation of cytoplasmic molecules. As evident from recent publications [115], this concept has the potential to revolutionize cell biology, however, it will be wise to learn from both progress and mistakes in the biomembrane field. The course of cytoplasmic/nuclear phase separation field seems extremely akin to the early 2000s when every protein was somehow found to be in membrane rafts. Despite the following dark years for raft concept, physical chemistry and biophysics of phase separation were extremely enlightening for membrane biology. Fundamental quantitative approaches of biophysics and physical chemistry will undoubtfully be essential tools for LLPS in nucleus/cytoplasm.

Awareness of methodological bias is another key aspect for reliable data. Therefore, studies showing the limits and bias of methods and techniques are very valuable for scientific progress. Detergent resistance assay, for instance, helped the field to realize the existence of more tightly-packed fractions within a membrane; however, it was not reliable enough to tell which components were exactly in this fraction. Each detergent type or concentration gave different results, which created unmatching lists of “rafty” proteins, and overall undermined the concept. Phase-separated GUVs also impressively demonstrated the physicochemical principles governing LLPS. However, due to the extreme nature of phases we discussed earlier, certain commonly used compositions are not suitable to infer phase partitioning of molecules in cell membranes. GPMVs are the best systems to study partitioning in a thermodynamical equilibrium yet physiologically relevant system. However, limitations of GPMVs are also shown [108,116,117].

Besides revealing the potentials and limitations of existing technologies, combining new technologies will also be instrumental to unravel the mysteries of biomembranes in the future. Lipidomics has recently gained popularity, and the most recent single-cell lipidomics approach has the potential to be a game-changer [118]. Super-resolution microscopy approaches and chemical biology tools are becoming constantly better. Since PM is the hub for almost all signaling events in cells, this area of research will always be full of interesting questions and, thanks to all the new technologies, full of exciting results. Research in upcoming years will pave the way towards unraveling the mystery of how potential phase separation in PM influences signaling events. Moreover, previously competing concepts for signaling at PM (such as clustering-based activation, size-based activation, phase separation, etc.) can be converged, and unifying concepts can be within our grasp.

## Figures and Tables

**Figure 1 membranes-11-00323-f001:**
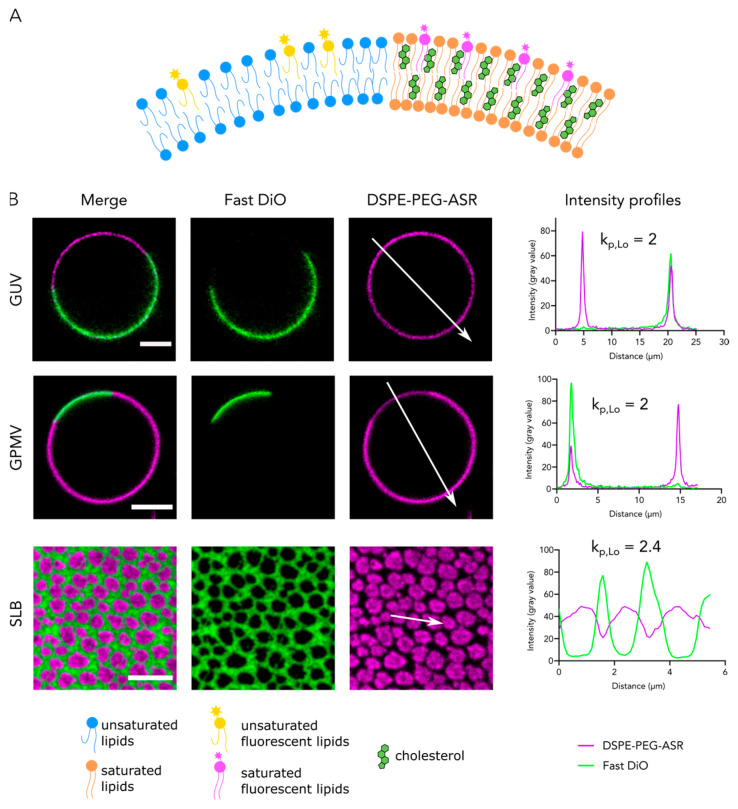
Visualization of phase separation in synthetic and cell-derived membrane systems using phase selective probes: (**A**) Cartoon of phase separation and distribution of phase-selective probes; (**B**) Confocal microscopy pictures of phase-separated GUVs, GPMVs, and SLBs. GUVs and SLBs are composed of the ternary mixture of DOPC/SM/Chol (2:2:1). GPMVs are extracted from HeLa cells. Fast DiO (green) incorporates preferentially in Ld domains whereas DSPE-PEG-ASR prefers Lo domains. Intensity line profiles were extracted along the white arrows as indicated. K_p,Lo_ is intensity in Lo phase divided by intensity in Ld phase. Scale bars are 5 µm.

**Figure 2 membranes-11-00323-f002:**
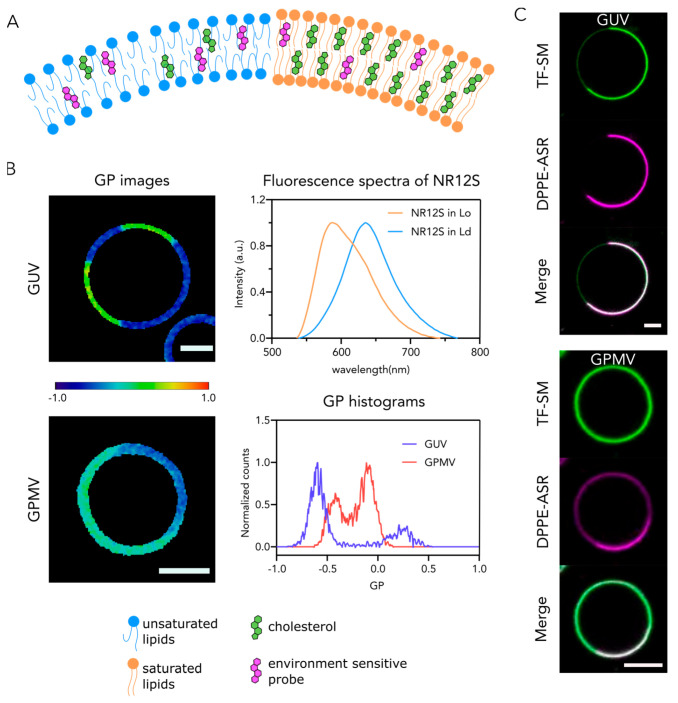
Membrane order in Lo and Ld phases of synthetic (GUVs) and cell-derived (GPMVs) membranes: (**A**,**B**) Visualization of phase separation in synthetic and cell-derived membrane systems using phase sensitive probes; (**A**) Cartoon of phase separation and distribution of phase- sensitive probes; (**B**) Ratiometric imaging of phase separation in GUVs and GPMVs using the environmental-sensitive probe NR12S. Fluorescence spectrum of NR12S exhibits a drastic red shift in Ld in comparison to Lo. The images of membranes at 560 nm and 650 nm emission wavelengths were recorded and used to produce GP images. The color code on the GP images corresponds to the color bar: dark red is +1, and dark blue is −1. The GP histograms for both GUV and GPMV show two distinct populations. In each case, low GP counts correspond to Ld whereas high GP counts correspond to Lo. Importantly, GP values of Ld and Lo are different for GUVs and GPMVs. (**C**) TopFluor-labeled SM (TF-SM, green) partitions into Ld in GUVs (DOPC:SM:Chol, 2:2:1) while it partitions to both phases in GPMVs derived from U2OS cells. The DPPE-Abberior Star Red (DPPE, ASR, magenta) partitions preferentially to Ld in both GUVs and GPMVs. Scale bars are 5 µm.

## Data Availability

The data presented in this study are openly available in FigShare at https://doi.org/10.3390/membranes11050323 (accessed on 28 April 2021).

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
