# Peer review of "How Does Liquid-Liquid Phase Separation in Model Membranes Reflect Cell Membrane Heterogeneity?"

_membranes, 2021, doi:10.3390/membranes11050323_

Round 1

Reviewer 1 Report

The review deals with the debated topic of liquid-liquid phase separation in model membranes and how reliable these models are in mimicking the behavior of native membranes. The manuscript contains an exhaustive description of liquid-liquid phase separation and of the model systems mainly used in this field. An adequate contextualization of the topic with references to lipid/membrane rafts has been provided, as well as the discussion on the lipid heterogeneity of the plasma membrane has been addressed. Attention was also paid to fluorescence microscopy as a crucial biophysical technique in the field, with particular reference to the molecular probes developed to study phase separation, giving rise to a section, in my opinion, particularly useful for organizing the enormous variety of molecular probes developed over the years. Biophysical properties of phase separation in the different model systems have been discussed as well, concluding that, although each model system own its advantages and disadvantages, from the point of view of phase separation, cell-derived GPMV are the best models as they preserve the original natural complexity.

In general, the manuscript is well-written, well-organized and properly referenced even if it can still be slightly improved. Figures are clear and appealing. Nevertheless, in some points the discussion can be broadened making reference to more recent works. Minor recommendations to improve the quality of the paper are listed below:

-when introducing the liquid-liquid phase separation, authors refer to the well-known case of cholesterol interacting with sphingomyelins; for completeness, recent findings about a very complex behavior of GUVs containing different kind of sphingomyelins which leads, at specific combinations of composition and temperature, to regions with three-phase coexistence should be considered as well as the role played by acyl chain mismatch on the formation of domains. The authors can critically use following papers: (https://doi.org/10.1016/j.bpj.2018.12.018;  https://doi.org/10.1016/j.bpj.2019.09.025)

- similarly, when dealing with fluorescent probes, some comments about the residues owning intrinsic fluorescence properties and their recent exploitation in fluorescent measurements should be added; this is the case of Kynurenine (Kyn 13) residue present in the Daptomycin structure, a commercial cyclic lipopetide used against Gram-positive bacteria, which has been recently demonstrated to strongly affect the phase behavior of model systems. The presence of Kyn residue coupled with DHPE-Texas Red allowed to prove that Daptomycin target PG lipids regardless their phase. Although this behavior has been analyzed on SLBs model systems, which do not contemplate the complexity of GPMVs, it represents a concrete example of interaction with phase-separated systems.

I believe that some comments on the aforementioned aspects may improve the review and provide a more complete overview. 

Author Response

We are grateful to the reviewers for the thoughtful comments. We addressed all their concerns in the revised manuscript. Below, we list the comments from the reviewers and the respective modifications to the manuscript. Below is point-by-point response.

-when introducing the liquid-liquid phase separation, authors refer to the well-known case of cholesterol interacting with sphingomyelins; for completeness, recent findings about a very complex behavior of GUVs containing different kind of sphingomyelins which leads, at specific combinations of composition and temperature, to regions with three-phase coexistence should be considered as well as the role played by acyl chain mismatch on the formation of domains. The authors can critically use following papers: (https://doi.org/10.1016/j.bpj.2018.12.018;  https://doi.org/10.1016/j.bpj.2019.09.025)

Indeed, more complex lipid compositions lead to variability of lipid domain properties (shapes, sizes, symmetry) in membranes that exhibit phase separation. In order to introduce this discussion into the review we added the following sentences in L148-156:

The necessity for LLPS in a lipid bilayer would be the presence of lipid species with different chain-melting temperatures55. Additionally,  lipid chain length mismatch contributes to the LLPS56,57. Notably, the geometrical and biophysical properties can be modulated by tuning parameters such as lipid composition, temperature, osmotic pressure, membrane tension etc58. As a result, in synthetic membranes (e.g. GUVs and SLBs) composed of the ternary mixture (e.g., of DOPC, SM and Chol, classical raft mixture), LLPS exists at specific lipid ratios15,16. Moreover, in extreme cases three phase coexistence can be observed, where in addition to Lo and Ld, “gel phase” domains appear14,59.

- similarly, when dealing with fluorescent probes, some comments about the residues owning intrinsic fluorescence properties and their recent exploitation in fluorescent measurements should be added; this is the case of Kynurenine (Kyn 13) residue present in the Daptomycin structure, a commercial cyclic lipopetide used against Gram-positive bacteria, which has been recently demonstrated to strongly affect the phase behavior of model systems. The presence of Kyn residue coupled with DHPE-Texas Red allowed to prove that Daptomycin target PG lipids regardless their phase. Although this behavior has been analyzed on SLBs model systems, which do not contemplate the complexity of GPMVs, it represents a concrete example of interaction with phase-separated systems.

We included the information regarding membrane-binding peptides in the manuscript by addition of the following sentence:

Furthermore, liquid phases can be visualized by proteins (such as lysenin, specific to sphingomyelin70) and charged peptides (such as daptomycin, specific to PG71) that specifically bind membrane lipids.”

Reviewer 2 Report

My comments are summarized in the attached pdf file.

Author Response

We are grateful to the reviewer for the thoughtful comments. We addressed all their concerns in the revised manuscript. Below, we list the comments from the reviewer and the respective modifications to the manuscript. In the revised manuscript, the new text is in red font.

… I could not understand the answer how liquid-liquid phase separation in model membrane systems reflect the heterogeneity of cell membranes. Since the behavior of cell membranes is very complicated, many studies have investigated some model systems. As the author mentioned in this review, it is dangerous to assume that the results obtained in the model systems are the same as those in the cell membranes. With this in mind, it is important to consider how to understand cell membranes through the model systems, and the authors should give some suggestions. I do not think the authors gave enough suggestions, and it is hard to say that they answered the question in the review title.

In the revised version, we tried to give more suggestions in the new section (5. LLPS in model membranes vs live cell PM heterogeneity). We are open to improve it further if the reviewer has specific suggestions.

The paragraph starting from L.122. Some studies tried to examine the behavior of pore-spanning membranes to eliminate the effects from the substrate (For example, see Sibold et al., Phys. Chem. Chem.Phys., 22, 9308-9315 (2020).). It is worth to mention such studies.

Indeed, pore-spanning SLBs are a good example of planar free-standing SLB. We added the sentence Furthermore, the deposition of the lipid bilayer on a porous surface allows for the production of so-called pore-spanning SLBs that offer studying of constrained planar regions of free-standing membranes43.

L.167-168, L.304, and L.306. It is not desirable to show only the abbreviations.

The full names of techniques and chemicals were introduced with abbreviations in brackets.

L.188 Show at least one specific cholesterol derivative.

The cholesterol derivatives (Top-fluor cholesterol, Cholestatrienol) were introduced in brackets.

Fig.1. I can see that the distribution of DSPE-PEG-ASR in GUV (and also GPMV) is almost uniform. As the authors wrote at L.216, \high partitioning coefficients will clearly distinguish phases.". So, this fluorescent probe is not suitable for an example showing the phase separation.

This is a very good point. In order to demonstrate the Lo preference of DSPE-PEG-ASR better, we added the intensity line profiles through the membranes to Figure 1. Moreover, we added kp,Lo values on figures. kp,Lo value is intensity in Lo divided by intensity in Ld for the line profiles. In the text, we also now give quantitative meaning to “high partitioning coefficients” by adding the expected kp,Lo (≥2) from a good Lo marker.

I cannot agree with this sentence, \phase separated GUVs are not good mimics of any possible heterogeneity in PM". Even if phase separated GUVs do not mimic the heterogeneity in PM well, I believe that we do not say that GUVs are not good. The important point is not good or bad. What is the difference between them? Why does such a difference occur? And, what can we understand from this difference? Such discussion leads to answer the question in the review title.

We agree that the sentence “GUVs are not good mimics of any possible heterogeneity in PM” is an inappropriate statement. We clarified that we mean the commonly used ternary mixture of DOPC, SM and cholesterol. Furthermore, in the next paragraph we added the possibility to modify phase behaviour by introduction of other phospholipids in the membrane mixture, which will bring the membrane order observed in GUVs closer to GPMV domains.

In addition, some readers may feel that all phase separated GUVs are not good for mimicking the PM heterogeneity. The simple systems like DOPC/SM/Chol are certainly not suitable. However, in a slightly more complicated system, the difference in order between Lo and Ld phases becomes smaller by the addition of lipid. For example, the difference of GP values between Lo and Ld phases becomes smaller by POPC (see Shimokawa et al., Phys. Chem. Chem. Phys., 17, 20882-20888 (2015).) and the phase separation behavior in DOPC/DSPC/POPC/Chol is significantly different from the the simple system DOPC/SM/Chol (see Konyakhina et al., Biophys. J., 20, L8-L10 (2011) and many relevant papers published from this group (G. W. Feigenson's group).). It is important to investigate whether the protein distribution in such a complicated system is the same as DOPC/SM/Chol.

We agree that it is important to mention that membrane order in lipid phases can be modulated by changing the lipid composition. Moreover, obviously replacing DOPC by POPC (for example) or sphingomyelin with DSPC (or DPPC) will modify the membrane order. We add these notions in the text with appropriate references. Furthermore, GUVs are widely used to reconstitute membrane protein partitioning into the lipid bilayer.

…This allows us to discuss whether protein distribution is simply determined by the order of the phases or not. I think that mentioning such points will increase the value of this review.

As far as we are aware, there is not any published information regarding protein partitioning in Lo domains composed of other ternary systems. Also, it is getting less popular to reconstitute proteins in ternary systems due to technical challenges. As we do not see this kind of experiments to be routinely done in the future, we did not add this discussion to the review.

Round 2

Reviewer 2 Report

The authors appropriately revised the manuscript and I am satisfied with the revisions.

I recommend that the current version is suitable for the publication in Membranes.